# A high-throughput Galectin-9 imaging assay for quantifying nanoparticle uptake, endosomal escape and functional RNA delivery

Michael J. Munson [1✉], Gwen O'Driscoll [1], Andreia M. Silva[2], Elisa Lázaro-Ibáñez [1], Audrey Gallud[3], John T. Wilson [4], Anna Collén[5], Elin K. Esbjörner [3] & Alan Sabirsh [1✉]

RNA-based therapies have great potential to treat many undruggable human diseases. However, their efficacy, in particular for mRNA, remains hampered by poor cellular delivery and limited endosomal escape. Development and optimisation of delivery vectors, such as lipid nanoparticles (LNPs), are impeded by limited screening methods to probe the intracellular processing of LNPs in sufficient detail. We have developed a high-throughput imaging-based endosomal escape assay utilising a Galectin-9 reporter and fluorescently labelled mRNA to probe correlations between nanoparticle-mediated uptake, endosomal escape frequency, and mRNA translation. Furthermore, this assay has been integrated within a screening platform for optimisation of lipid nanoparticle formulations. We show that Galectin-9 recruitment is a robust, quantitative reporter of endosomal escape events induced by different mRNA delivery nanoparticles and small molecules. We identify nanoparticles with superior escape properties and demonstrate cell line variances in endosomal escape response, highlighting the need for fine-tuning of delivery formulations for specific applications.

[1] Advanced Drug Delivery, Pharmaceutical Sciences, BioPharmaceuticals R&D, AstraZeneca, Gothenburg, Sweden. [2] Discovery Biology, Discovery Sciences, BioPharmaceuticals R&D, AstraZeneca, Gothenburg, Sweden. [3] Division of Chemical and Biomolecular Engineering, Department of Biology and Biological Engineering, Chalmers University of Technology, Gothenburg, Sweden. [4] Department of Chemical and Biomolecular Engineering, Vanderbilt University, Nashville, TN, USA. [5] Projects, Research and Early Development, Cardiovascular, Renal and Metabolism, Biopharmaceuticals R&D, AstraZeneca, Gothenburg, Sweden. ✉email: Michael.Munson@astrazeneca.com; Alan.Sabirsh@astrazeneca.com

Oligonucleotide therapies (including siRNA, antisense oligos (ASOs) and mRNA) are therapeutic interventions that target or deliver protein-encoding mRNA; these can therefore modulate, with high specificity, targets that are considered undruggable by classical small-molecule approaches or other traditional modalities[1,2]. Hydrophilic oligonucleotides cannot be readily taken up across cellular membranes and are prone to degradation by endogenous nucleases, therefore the use of delivery vectors for both protection and delivery is required[3]. Delivery systems such as lipid nanoparticles (LNPs) or polymeric nanoparticles (PNPs) are being explored in an attempt to bolster the intracellular delivery of oligonucleotides, however, these systems still have disadvantages concerning safety, stability and most importantly delivery efficacy[4]. Despite progress in the development and design of LNP and PNP vectors[5], the delivery efficacy of short oligonucleotides using synthetic nanoparticles remains poor. It is estimated that only 1.5–3.5% of siRNA cargo taken up by cells reaches the cytoplasm, with approximately half of the nucleotide cargo released when an endosomal disruption event does occur[6,7]. Oligonucleotides such as mRNA are considerably larger (~600–10,000 kDa) than ASOs/siRNA (~4–14 kDa), likely diminishing delivery yields even further[5]. Endosomal escape is therefore one of the most substantial bottlenecks for successful oligonucleotide delivery. Improving endosomal release is a key driver for delivery system design and will be of great importance for the further development of oligonucleotide therapeutics, particularly for treatments that require high therapeutic protein levels.

LNPs are of particular interest as delivery vectors due to their relative stability, low toxicity and ease of large-scale preparation[8]. However, optimising the chemical and physical features of these entities to improve functional delivery is a complex task. LNPs are typically comprised of four major lipid components: an ionisable lipid, a sterol, a phospholipid and a lipidated polyethylene glycol (PEG). The ionisable lipid has been the primary subject of development with combinatorial libraries often used to identify new lipids[9,10]. However, the undeniable importance of other components such as the sterol or the PEG-lipid for modulating uptake, improving pharmacokinetics or modifying the particle's corona composition is being increasingly appreciated[11–14]. Optimising the relative abundance of each component can further modulate potency by up to sevenfold[15], but to fully explore all of these parameters requires fast and robust screening assays. Despite the central importance of endosomal escape, there are currently limited established options to evaluate this event in living cells[16].

Members of the Galectin (GAL/LGALS) family of proteins have been exploited as reporters of endosomal escape in a variety of contexts. Galectins are primarily expressed in the cytosol and contain carbohydrate recognition domains (CRDs) that bind to β-galactoside sugars[17]. Galectins can be recruited to endosomes when membrane damage exposes β-galactosides on the inner leaflet of the endosomal membrane to the cytosol[18]. Point mutations that strongly reduce the binding affinity of the CRD also prevent endosomal relocalisation of GAL3[18–20]. The exposure of β-galactosides often occurs during host-pathogen interactions as a method of identifying damaged cellular membranes and aims to assist in the clearance of pathogens[21,22]. GAL8/9 are also robustly recruited to sites of endosomal damage in response to artificial delivery vehicles such as LNPs or PNPs that induce endosomal escape as a mechanism of nucleotide cargo delivery[7,23]. Endosomal escape induced localisation of GAL8 was shown to recruit autophagy adaptors as part of a clearance response against *Salmonella* and therefore forms an important part of cellular host-defence[22]. GAL9 recruitment has no similarly reported perturbing effects on normal cellular function and was recently reported to evoke a greater recruitment response to endolysosome damage than GAL3 or GAL8[24].

To address the basic limitations of characterising nanoparticle function, we have developed a GAL9 based endosomal escape imaging assay to allow time-lapse live-cell recordings of endosomal damage events. In combination with fluorescently labelled nanoparticles, this single assay allows a complete overview of nanoparticle trafficking from cellular uptake to endosomal escape and ultimately mRNA translation to functional protein whilst also evaluating potential cellular toxicity within a high-throughput format. We first demonstrate that this assay has a large signal window for the detection of endosomal escape events, confirming its suitability as a screening assay. We subsequently validate GAL9 recruitment in response to cell treatments with small molecules, LNPs and PNPs and highlight cell line-specific differences in sensitivity to endosomal escape. Finally, we demonstrate that this assay can be combined with robotic LNP formulation, allowing extensive and rapid screening of nanoparticle formulations across multiple, biologically relevant parameters.

## Results

**Generation of mCherry-Galectin9 reporter cells**. We generated four stable mCherry-GAL9 reporter expression cell lines using the ObLiGaRe zinc-finger nuclease (ZFN) knock-in strategy[25]. Cell lines were transiently transfected with an *AAVS1*-targeting ZFN plasmid and a donor construct encoding a puromycin resistance gene and mCherry-GAL9 under the E1α promoter (Fig. 1a). Positively integrated cells were selected with puromycin and flow-sorted into pools with similar mCherry-GAL9 expression levels (Supplementary Fig. 1a, b). Gene copy number analysis of HepG2 cells revealed on average ~2.3 integrated copies (Supplementary Fig. 2) of the mCherry-GAL9 reporter compared to HeLa (2 copies), NCI-H358 (2.9 copies) or Huh7 (3.5 copies) (Fig. 1b). The integration site was verified using primers to amplify between the *AAVS1* locus and the reporter sequence, yielding the expected 251 bp product (Fig. 1c). Cell expression of the mCherry-GAL9 protein was verified by western blotting using an α-mCherry antibody for detection (Fig. 1d). Two prominent bands were observed, corresponding to full-length (~70 kDa) and truncated (~50 kDa) mCherry-GAL9 (consistent with the loss of GAL9's C-terminal CRD (~20 kDa) due to protease cleavage[26]). Cells modified to express mCherry-GAL9 exhibited generally diffuse cytosolic mCherry staining under standard growth conditions (Fig. 1e). Comparison of the cellular mCherry fluorescence intensity by microscopy (Fig. 1f) supported the observations by western blot that HepG2 and NCI-H358 cells expressed higher levels of protein than HeLa and Huh7 cells (Fig. 1d, f). This suggests that cell-lineage related differences in reporter protein abundance exist that are not directly due to copy number variation but may arise from differential regulation of galectin trafficking and secretion[27]. The protein abundance, however, was not decisive for assay function (see following).

**mCherry-GAL9 is recruited to small molecule induced damage**. To validate that mCherry-GAL9 can be recruited to sites of endolysosome damage, we exposed Huh7 mCherry-GAL9 cells to small molecules known to induce endomembrane disruption (Fig. 2a). We first used chloroquine, a cationic amphiphilic drug (CAD) that accumulates in late endosomal compartments and was recently used by Du Rietz et al. to study endosome rupture in GFP-GAL9 expressing HeLa cells[24,28]. Following administration of chloroquine, we observed dose-dependent induction of large mCherry-GAL9 positive structures with a maximal response after 8–12 h of treatment and at ~60–80 μM, consistent with the

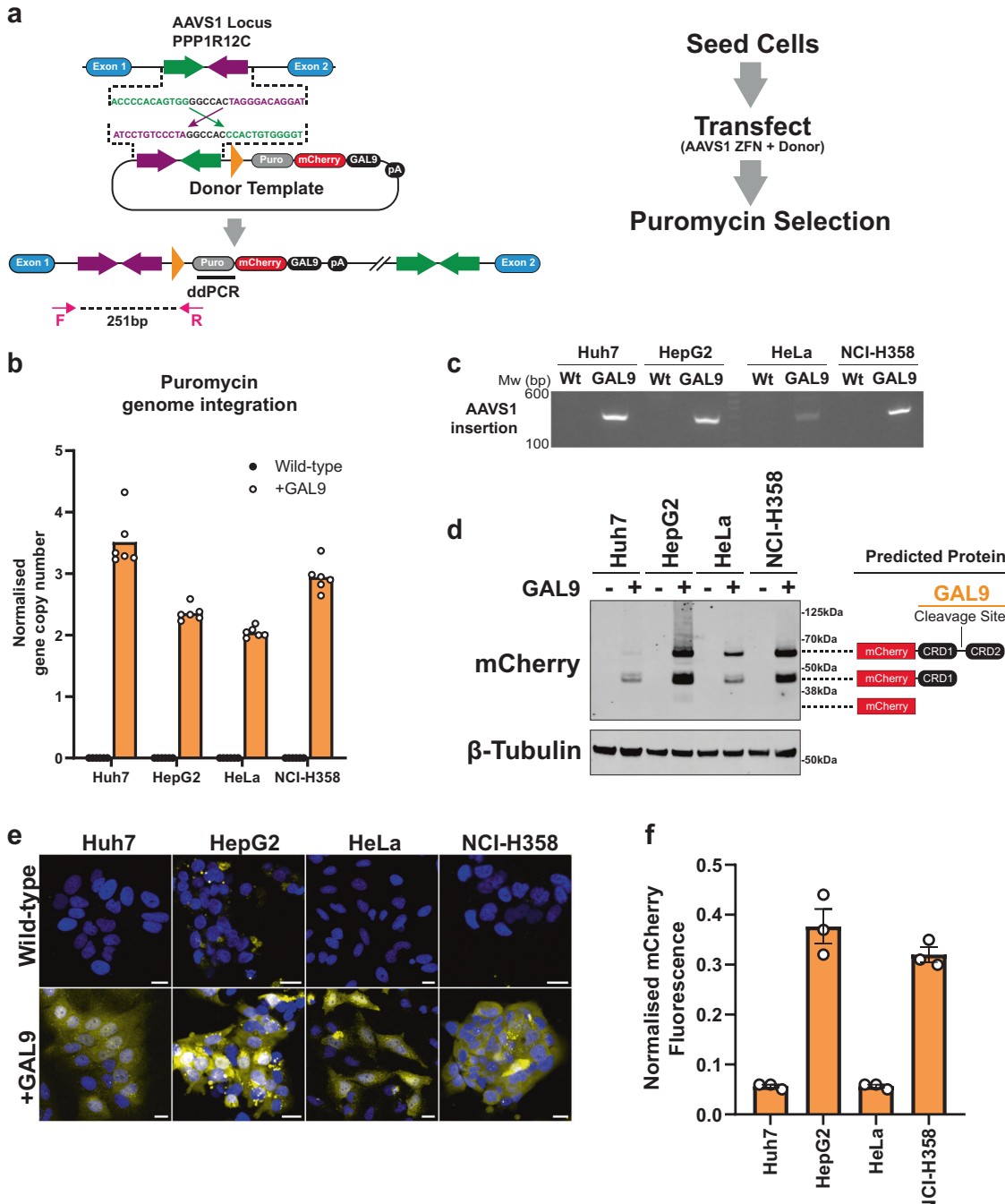

**Fig. 1 Generation of mCherry-GAL9 knock-in cell lines. a** Knock-In strategy for mCherry-GAL9. Reporter plasmid contains analogous zinc finger nuclease (ZFN) sites to *AAVS1* locus. Co-transfection with an *AAVS1* targeted ZFN induces a double-strand break and a non-homologous end-joining repair mechanism, resulting in mCherry-GAL9 insertion under an EF1α promoter with a puromycin selection cassette. **b** Average copy number of puromycin genomic insertions per cell determined by droplet digital PCR from $n = 6$ replicates, normalised to the reference gene AP3B1, as an indirect measurement of mCherry-GAL9 insertions. **c** Representative agarose gel of the amplicon spanning the AASV1 locus and the donor mCherry-GAL9 donor vector, confirming the insertion of the construct at the expected genomic location. Molecular weight (base-pairs, bp) of standards is indicated next to the gel. **d** Western blot of stably integrated mCherry-GAL9 cell lines for indicated proteins and non-transfected cell controls, 40 µg per lane. **e** Fluorescence images of Huh7, HepG2, HeLa and NCI-H358 cells ± insertion of the mCherry-GAL9 reporter under normal growth conditions, scale bar = 20 µm. Note: LUTs between cell lines vary (± mCherry-GAL9 reporter are matched), this is intended for clarity of GAL9 signal within cells. See (**f**) for true fluorescent intensity. **f** Quantitation of cellular mCherry fluorescence intensity from reporter lines. Values are normalised means from $n = 3$ independent experiments ± SEM.

positive response obtained by Du Rietz et al. (Fig. 2b, c)[24]. We then tested three small molecules from a structurally related family (UNC10217938A, UNC2383, and UNC4267 - Fig. 2a, b) which were previously identified to enhance splice-switching oligonucleotide and ASO delivery by inducing endomembrane permeabilisation[29,30]. The response to UNC compounds was rapid with maximal mCherry-GAL9 recruitment occurring at 1–3 h post-dosing (Fig. 2c); furthermore, the effect was seen at a much lower dose (0–15 µM) compared to chloroquine. UNC2383 was the most potent compound with the induction of large

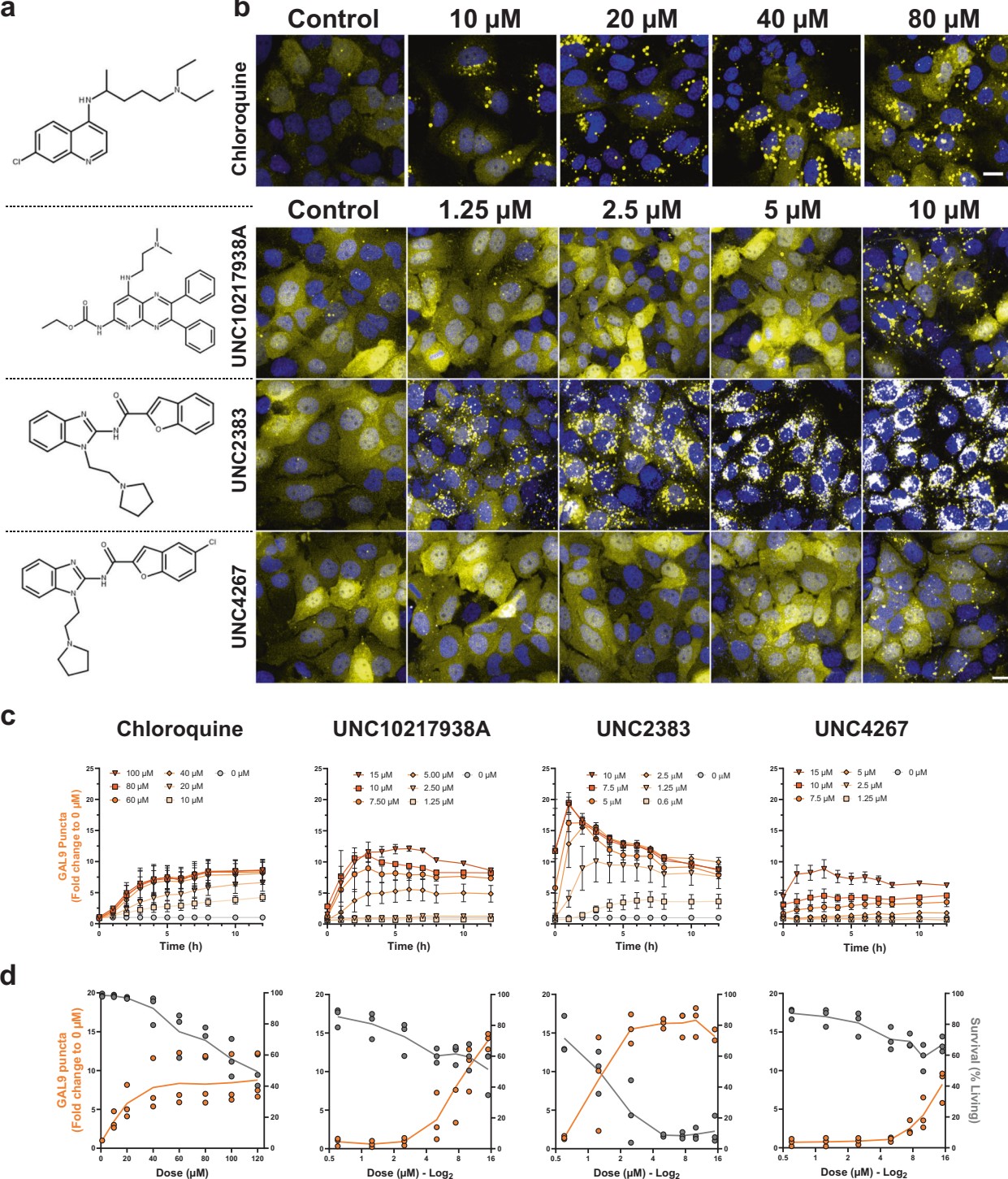

**Fig. 2 Small Molecules induce mCherry-GAL9 recruitment. a** Chemical structures of chloroquine and UNC compounds **b** Representative images of Huh7 mCherry-GAL9 cells following 2 h incubation with indicated small molecules and doses, scale bar = 20 μm. **c** Kinetic analysis of mCherry-GAL9 puncta formed across 0–12 h of incubation with indicated compounds and doses, values represent normalised means ± SEM from n = 3 independent experiments. **d** Comparison of mCherry-GAL9 puncta induced (Orange line - 8 h chloroquine, 2 h UNC compounds) to cell survival (Grey line - 12 h post-dosing, determined by Hoechst morphology) relative to the compound dose in Huh7 mCherry-GAL9 cells. Lines represent mean values from n = 3 independent experiments.

numbers of small mCherry-GAL9 structures from 1.25 μM (Fig. 2d). Our observations match both the timescale and the required doses to achieve functional improvement of oligonucleotide delivery reported previously for UNC2383 and supports that this effect is indeed functionally linked to endosome damage[31]. Simultaneously, we inferred cellular toxicity from nuclear condensation that leads to smaller and more intense Hoechst stained nuclear structures[32]. All compounds tested elicited partial cellular toxicity that correlated with the level of GAL9 recruitment (Fig. 2d). The ability to utilise nuclear

morphology and intensity to infer toxicity for simultaneous evaluation of compound safety is useful for future screening of compound libraries to identify novel small molecules that can induce endosomal escape with improved safety profiles.

Our data demonstrate that the mCherry-GAL9 recruitment assay can sensitively detect differences in the endosomal damaging potencies of small molecule compounds and that their effect does not require the presence of oligonucleotides. We next determined the robustness of the mCherry-GAL9 recruitment assay in Huh7 cells by calculating the Z' factor in response to UNC2383 treatment at 10 μM. The Z' Factor is a measure of separation between the minimum (vehicle) and maximum response (UNC2383 10 μM) often used in the development of screening assays[33]. The Z' factor for Huh7 cells was calculated as +0.76; assays rated above +0.5 are classified as excellent screening assays[33]. This validates that a large signal window exists and that the mCherry-GAL9 reporter line enables robust detection of endosomal damage.

**mCherry-GAL9 recruitment in response to mRNA delivery using LNPs.** The reporter cell lines were exposed to LNPs formulated by microfluidic mixing of Cy5-labelled, EGFP-encoding mRNA with lipids, using the ionisable lipid DLin-MC3-DMA (MC3) to enable delivery[9]. Uptake of MC3-LNPs was observed as a punctate Cy5 signal within wild-type and mCherry-GAL9 cells, as demonstrated after 3 h of incubation with Huh7 cells (Fig. 3a). Quantification showed that there was no significant difference in the number of intracellular Cy5-mRNA puncta between wild-type and mCherry-GAL9 expressing cells (Fig. 3b), indicating that the introduction of the mCherry-GAL9 gene did not affect cell uptake. Considerable relocalisation of the initially diffuse cellular mCherry-GAL9 fluorescence into bright punctate structures was observed following MC3-LNP uptake (Fig. 3a, b) and these often co-occurred with the Cy5-mRNA signal (Fig. 3a). This indicates that the MC3-LNPs induce exposure of β-galactoside ligands to the cytosol permitting mCherry-GAL9 binding. EGFP fluorescence, indicative of successful cytosolic mRNA delivery, could also be observed from 3 h (Fig. 3a, b). No differences were observed in EGFP intensity between wild-type and mCherry-GAL9 cells after 12 h (Fig. 3b). Further comparison of MC3-LNP uptake and EGFP expression in wild-type and mCherry-GAL9 Huh7 cells across time (0–14 h post-dosing) and across a dose-range (0.1–1 μg/ml) (Supplementary Fig. 3a, b) further emphasised that mCherry-GAL9 integration did not significantly alter the normal cellular processing (uptake and mRNA delivery) of the LNPs. This indicates that mCherry-GAL9 recruitment can be used to assess endosomal escape without significantly perturbing the normal trafficking of LNPs.

We next examined whether the formation of mCherry-GAL9 puncta correlates with the levels of LNP exposure and uptake, by testing a range of MC3-LNP doses across all reporter cell lines, and as a function of incubation time, by high-content live-cell microscopy (Fig. 3c, d). Dose-dependent increases in Cy5-mRNA structures, induction of mCherry-GAL9 structures and EGFP formation were observed as exemplified in Fig. 3c for HeLa mCherry-GAL9 cells after 10 h of exposure. The Cy5-mRNA and mCherry-GAL9 punctate structures were quantified over 0–14 h across all reporter lines (Fig. 3d). Interestingly, the four cell lines displayed distinct differences in both the degree and kinetics of MC3-LNP uptake, which was also reflected in the levels of endosomal escape. Furthermore, the cells demonstrated variable transfection with liver lines Huh7 and HepG2 producing the highest levels of EGFP protein (Fig. 3d, Supplementary Fig. 4a). HepG2 cells had a notably higher level

of background autofluorescent structures in mCherry and Cy5 channels, however, LNP-induced changes were still observable above this background level (Fig. 3d). We observed a delayed onset and peak of Cy5 and GAL9 puncta formation between 4 and 10 h in cell lines following LNP addition, this likely represents the time required for efficient PEG-shedding and/or protein corona formation in the cell culture media. These changes are prerequisites for efficient particle uptake.

Comparison of reporter lines demonstrated that Huh7 cells are the most sensitive to endosomal escape with mCherry-GAL9 recruitment from 0.02 μg/ml and an EC50 of 0.1 μg/ml, ~2–4 fold lower than other cell lines (Fig. 3e, Supplementary Table 1). Correlations between Cy5-mRNA puncta and mCherry-GAL9 structures (Fig. 3f) were explored using linear regression, revealing differences between cell lines both with respect to slope and goodness of fit ($R^2 = 0.573$–$0.9188$), indicating that multiple factors likely exist in relating the level of endosomal escape to particle uptake (Fig. 3f, Supplementary Table 1). In general, liver cell lines evoked higher endosomal escape for equivalent levels of MC3-LNP uptake (Fig. 3f). Comparison of MC3-LNP uptake to dose demonstrated less obvious differences between cell lines (Supplementary Fig. 4b), however, liver cell lines exhibit lower EC50 values than the other reporter lines and this may indicate higher efficiency MC3-LNP uptake at lower particle concentrations (Supplementary Table 1), possibly driven by receptor-mediated endocytosis[34].

Huh7, HepG2 and HeLa cells have very high levels of transfection in response to MC3-LNPs, while NCI-H358 cells had notably fewer EGFP positive cells (~50% at 0.2 μg/ml). NCI-H358 cells often grow in clusters in standard culture conditions and we observed that the outer 'edge' cells had higher LNP uptake, substantial relocalisation of mCherry-GAL9, and therefore preferentially expressed EGFP (Fig. 3g). By contrast, the MC3-LNPs appeared incapable at transfecting NCI-H358 cells in the 'core' of cell clusters. This result demonstrates that the mCherry-GAL9 recruitment assay selectively identifies individual cells undergoing endosomal escape and delivery of mRNA to the cytosol for translation, enabling analysis of cell-to-cell variation.

We next explored the use of small molecule kinase inhibitors targeting mTOR, a well-established regulatory kinase of protein synthesis, cellular growth and a negative regulator of autophagy[35]. We wanted to validate that manipulating cellular translation rate would also impact the translation of exogenous mRNA delivered by LNPs, verifying that our reporter line can demonstrate differences in the cellular translation rate. However, macroautophagy induced by mTOR inhibition or stress responses could potentially deliver cytosolic material such as our reporter to the lysosome, leading to a false positive signal[35]. We found no significant formation of mCherry-GAL9 structures upon induction of autophagy using dual mTOR complex (mTORC1/2) inhibitors, Torin1 or KU0063794 (Supplementary Fig. 5a–c) indicating autophagy induction does not lead to a false positive signal[36,37].

Co-incubation of MC3-LNPs with Torin1, KU0063794 or the mTORC1 specific inhibitor rapamycin led to a reduction in EGFP expression in Huh7 cells, without visibly altering Cy5-mRNA uptake or mCherry-GAL9 recruitment (Supplementary Fig. 6a). Quantification revealed up to a 60% reduction in EGFP translation with Torin1 treatment compared to vehicle controls, while no significant change was seen in particle uptake or endosomal escape indicating the difference was due to reduced mRNA translation (Fig. 3h). This reduction was observed with all mTOR inhibitors (Supplementary Fig. 6b). Importantly, this confirms that variations in protein translation rate are observable using our mCherry-GAL9 reporter cells and that this can occur independently of endosomal escape.

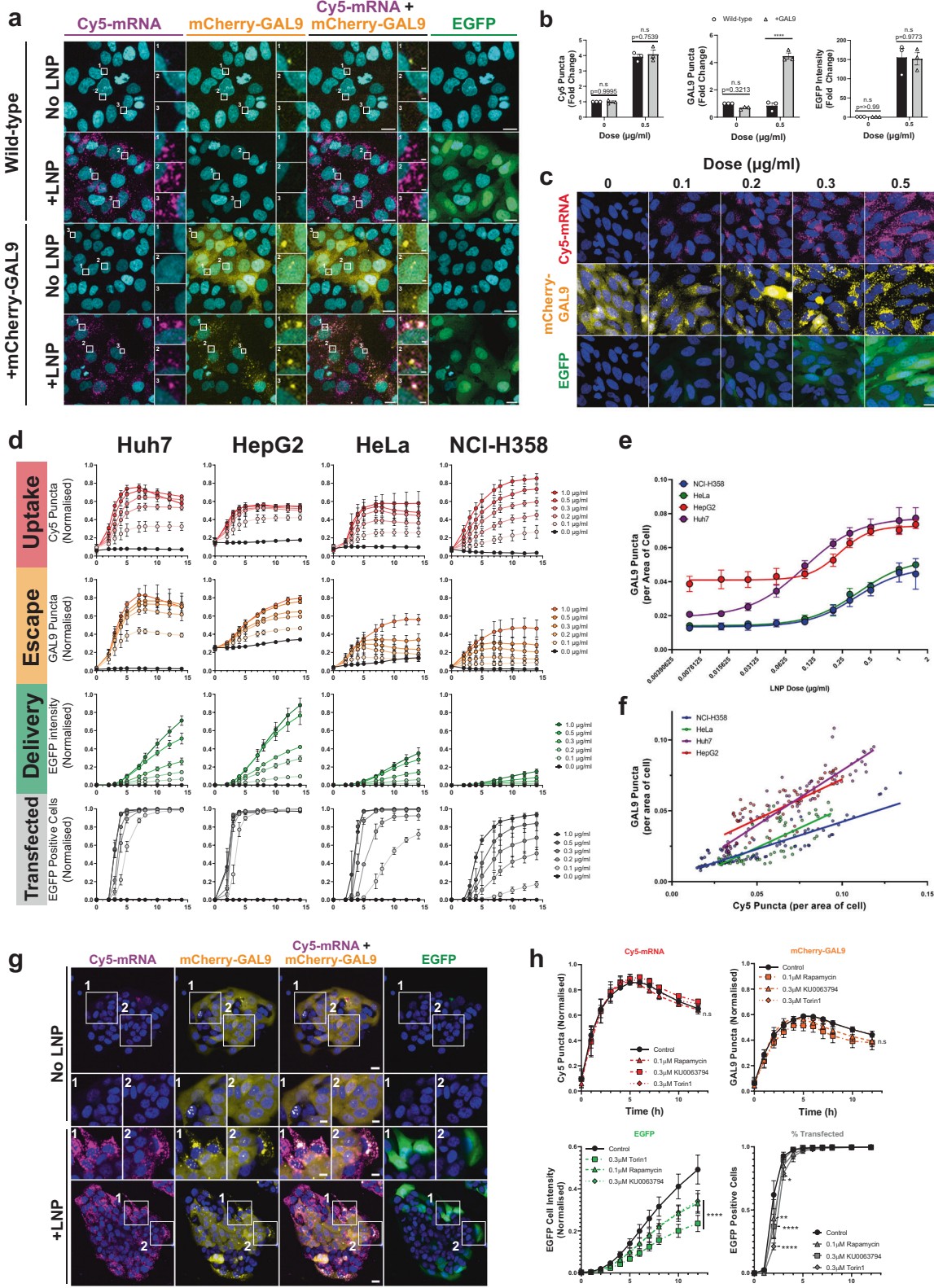

**PNPs induce mCherry-GAL9 recruitment**. To demonstrate the broad applicability of the mCherry-GAL9 recruitment assay we examined a different nucleotide delivery system based on an ionisable polymer series formed from DEAEMA (diethylaminoethyl methacrylate) and BMA (butyl methacrylate). This series varies in molecular weights but with a maintained ratio of 60:40 of the two moieties, with or without a $PEG_{2k}$ first block (Fig. 4a)[38,39].

We dosed mCherry-GAL9 HeLa cells with EGFP-encoding mRNA formulated with the different polymers and monitored for induction of endosomal escape by mCherry-GAL9 relocalisation and EGFP production (Fig. 4b–d). The polymers were able to induce endosomal escape, but the response decreased with increasing molecular weight (Fig. 4c). This correlated with the level of EGFP expression observed (Fig. 4d). The addition of a

**Fig. 3 LNPs induce mCherry-GAL9 recruitment. a** Huh7 cells ± mCherry-GAL9 reporter were dosed with 0.5 µg/ml MC3-LNPs for 3 h and imaged live, scale bar = 20 µm, insets scale bar = 2 µm. **b** Quantitation of mCherry-GAL9 and Cy5-mRNA objects (3 h post-dosing) along with cellular EGFP intensity (12 h post-dosing) shown as fold-change to the untreated wild-type control from $n = 3$ independent experiments ± SEM. **c** mCherry-GAL9 HeLa cells were treated with a dose range of MC3-LNPs (0–0.5 µg/ml) and imaged using live-cell microscopy at 10 h post-dosing, scale bar = 20 µm. **d** Quantitation of dose response in Huh7, HepG2, HeLa or NCI-H358 cells containing mCherry-GAL9 reporter across 0–14 h using 0–1 µg/ml MC3-LNPs. Values indicate LNP uptake (Cy5-mRNA structures), endosomal escape (mCherry-GAL9 structures), mRNA delivery (EGFP fluorescence) and transfection (% EGFP positive cells). Values were normalised to 0–1 (min-max) per replicate and combined to represent normalised means ± SEM from $n = 3$ independent experiments. **e** Sum of GAL9 puncta per LNP dose across time course obtained in (**d**) was plotted against LNP dose per cell line ± SEM from $n = 3$ independent experiments. Four-parameter logistic curve with variable slope fitted for each cell line to mean values to obtain $EC_{50}$ of LNP dose for GAL9 puncta induction. **f** Cy5 puncta were plotted against mCherry-GAL9 puncta across time course obtained in (**d**) per cell line, linear regression was fitted for each. Values obtained and fits for **e**, **f** are shown in Supplementary Table 1. **g** mCherry-GAL9 NCI-H358 cells dosed with 0.5 µg/ml MC3 and imaged after 10 h of incubation, scale bar = 20 µm, insets = 10 µm. **h** mCherry-GAL9 Huh7 cells co-dosed with 0.5 µg/ml MC3-LNP and 0.3 µM Torin1, 0.3 µM KU0063794, 0.1 µM Rapamycin or DMSO control. Quantitation of indicated structures/cell intensity. Values represent normalised means from $n = 3$ independent experiments ± SEM. Significance was determined by two-way ANOVA followed by Dunnett's post-test to the DMSO control where *$p < 0.05$, **$p < 0.01$, ***$p < 0.001$, ****$p < 0.0001$ and n.s. not significant.

PEG$_{2k}$ leader sequence led to a block in endosomal escape and a consequential lack of EGFP expression (Fig. 4b–d). Examination of effective PNPs demonstrated that the mCherry-GAL9 structures formed overlapped with Cy5-mRNA structures (Fig. 4e), similar to what was observed for LNPs (Fig. 3a). Particles formed from polymers lacking PEG$_{2k}$ were very large in size compared to LNPs, with hydrodynamic diameters of >500 nm according to DLS measurements (Supplementary Table 2). The large particle size led to relatively few particles being taken up per cell and consequently allowed comparatively simple real-time imaging of the intracellular trafficking of individual Cy5-mRNA containing structures over time, allowing us to capture single escape events. Capturing the trajectory of one single PNP (Fig. 4f), we observe its uptake into a cell and found that after 15–30 min, the Cy5-mRNA puncta became mCherry-GAL9 positive, indicating that endosomal disruption has occurred. EGFP translation and cellular fluorescence was observed after ~1 h, confirming functional delivery of the EGFP mRNA cargo. Therefore, we are remarkably able to resolve mCherry-GAL9 recruitment and mRNA translation down to a single endosomal release event.

**Endosomal escape is enhanced when LNPs contain β-Sitosterol.** Having established the mCherry-GAL9 recruitment assay and characterised its functional response to CADs, LNPs, and PNPs, we returned to investigations of how the lipid composition of LNPs influences their potency. It was recently shown that MC3-LNPs formulated with β-sitosterol instead of cholesterol have enhanced functional delivery of luciferase mRNA due to suspected structural changes that enhance fusogenicity[12,40]. We therefore compared MC3-LNPs formulated with either cholesterol or β-sitosterol and examined their uptake, ability to induce mCherry-GAL9 recruitment and functional delivery (i.e. EGFP expression) across cell lines.

Whilst the Cy5-mRNA uptake was similar between the two LNP types, the formation of mCherry-GAL9 positive structures and subsequent EGFP expression was markedly increased in all reporter lines following the substitution of cholesterol to β-sitosterol (Fig. 5a). In Huh7, HeLa, and HepG2 cells, the effect was most notable at the lower dose range (0.1–0.5 µg/ml) because of an eventual saturation of the EGFP translation response, which occurred at much lower concentrations for β-sitosterol compared to cholesterol MC3-LNPs (e.g. 0.3 µg/ml vs 1 µg/ml in HeLa cells) (Fig. 5b). For NCI-H358 cells, the positive effect of β-sitosterol was observed across the full dose range (Fig. 5b). Time course analysis over a period of 0–14 h was used to compare cholesterol and β-sitosterol MC3-LNPs delivered at a dose of 0.1 µg/ml (Huh7, HeLa, HepG2) or 1 µg/ml (NCI-H358) (Fig. 5c). Cy5-mRNA uptake was similar or mildly improved (HepG2)

whilst substantial increases in the number of mCherry-GAL9 structures upon β-sitosterol substitution were observed in all cell lines with concomitant differences in the resulting EGFP expression (Fig. 5c). β-sitosterol invoked mCherry-GAL9 structures across a longer time period than cholesterol particles (HeLa/HepG2: β-sitosterol plateau at 8 h, cholesterol at 3 h). Higher doses of LNPs (1 µg/ml) resulted in a similar mCherry-GAL9 response in Huh7, HeLa and HepG2 cells and this in turn yielded similar EGFP production (Supplementary Fig. 7). NCI-H358 cells, however, still benefited from the substitution to β-sitosterol at the highest dose of 1.5 µg/ml. Comparison of Cy5-mRNA structures against mCherry-GAL9 structures across doses in NCI-H358 cells confirmed that β-sitosterol induced higher quantities of endosomal escape for equivalent particle uptake (Fig. 5d), consistent with their suggested increased fusogenicity[12].

Taken together, these data show that LNP variants can have similar levels of uptake, yet the levels of endosomal escape that the LNPs elicit inside of cells can function independently of uptake, leading to differences in subsequent mRNA delivery and protein translation.

**Integration of the mCherry-GAL9 reporter lines into an LNP formulation screening platform.** Throughout this study we have utilised MC3:Cholesterol:DSPC:DMPE-PEG LNPs to deliver mRNA. This LNP formulation is well-characterised and performs well in vitro and in vivo and is a suitable tool delivery vehicle to evaluate new assay systems[6,9]. However, identification of novel particle formulations that outperform this benchmark are needed for broad applicability of mRNA therapeutics in vivo. We established a screening platform, integrating the use of a robotic LNP formulation method, high-throughput particle characterisation (for LNP size and encapsulation efficiency) and dosing of reporter cells for live-cell imaging (Fig. 6a). We generated a panel of 72 LNP formulations as a proof of concept, containing one of three ionisable lipids (DLin-DMA, DLin-KC2-DMA, DLin-MC3-DMA—hereafter denoted DMA/KC2/MC3)[41,42], four phospholipids (DSPC, DOPC, DSPE, DOPE), three sterols (Cholesterol, 7β-Hydroxycholesterol, 25-Hydroxycholesterol—hereafter 7B-HC/25-HC), and two PEG-lipids (DMPE-PEG, DMG-PEG) (Supplementary Fig. 8a), including the standard MC3-LNP formulation that has been used throughout this study.

LNPs formed with the robotic system were larger at 150–400 nm (Supplementary Fig. 8b), compared to the LNPs used in previous experiments (which were formulated using a microfluidic device). Their size was determined using a DLS plate reader with 90° scattering angle, which may overstate actual particle size[43]. Size measurement of the same LNPs using a

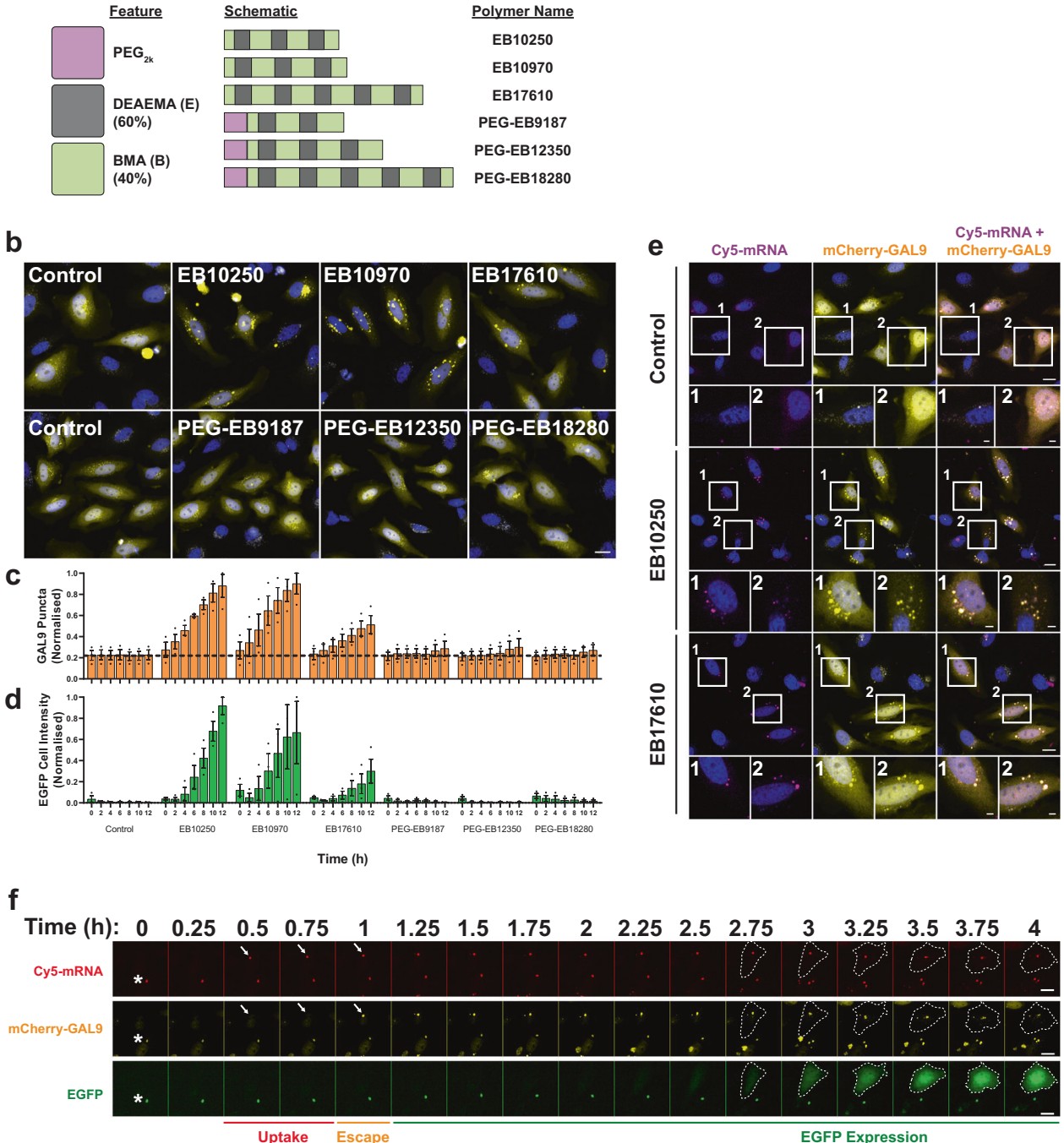

**Fig. 4 mCherry-GAL9 responds to polymeric nanoparticles. a** Overview of polymer names and compositions. Polymers are comprised of DEAEMA (60%) and BMA (40%) at variable lengths ± PEG₂ₖ first block. **b** Representative images of HeLa mCherry-GAL9 cells dosed with 1 µg/ml of indicated polymers in serum-free media and imaged by live-cell microscopy, scale bar = 20 µm. **c** Quantitation of cells treated as in (**b**) for mean mCherry-GAL9 structures or **d** mean EGFP expression both ± SEM from n = 3 independent experiments. **e** Representative images of HeLa mCherry-GAL9 cells treated with 1 µg/ml of EB10250, EB17610 or PBS control. Scale bar = 20 µm, insets = 5 µm. **f** Images of HeLa mCherry-GAL9 cells dosed with EB10250 polymer at 1 µg/ml in serum-free media and imaged live. Arrows indicate a single Cy5-mRNA particle taken up by a HeLa cell and undergoing endosomal escape, scale bar = 20 µm. Asterix indicates a non-specific autofluorescent signal present in all channels.

Malvern Zetasizer cuvette-based instrument with 173° scattering angle yielded Z-average sizes ~25% smaller (Supplementary Fig. 8c) confirming this view, but LNPs were nevertheless larger than NanoAssemblr formed LNPs. In addition, robotic LNP formulation resulted in poorer mRNA encapsulation (30–60%) than NanoAssemblr formed particles (~95–99%) (Supplementary Fig. 8b). Using this screening platform, we exposed mCherry-GAL9 Huh7 cells to LNPs and monitored particle uptake, mCherry-GAL9 recruitment, EGFP expression and percentage of cellular transfection by time-lapse microscopy as established earlier (Fig. 3d). For easier comparison between LNP formulations, a heatmap was generated for each assayed parameter (Cy5-mRNA uptake, GAL9 recruitment, EGFP expression, percent transfection) at 14 h post-dosing (Fig. 6b).

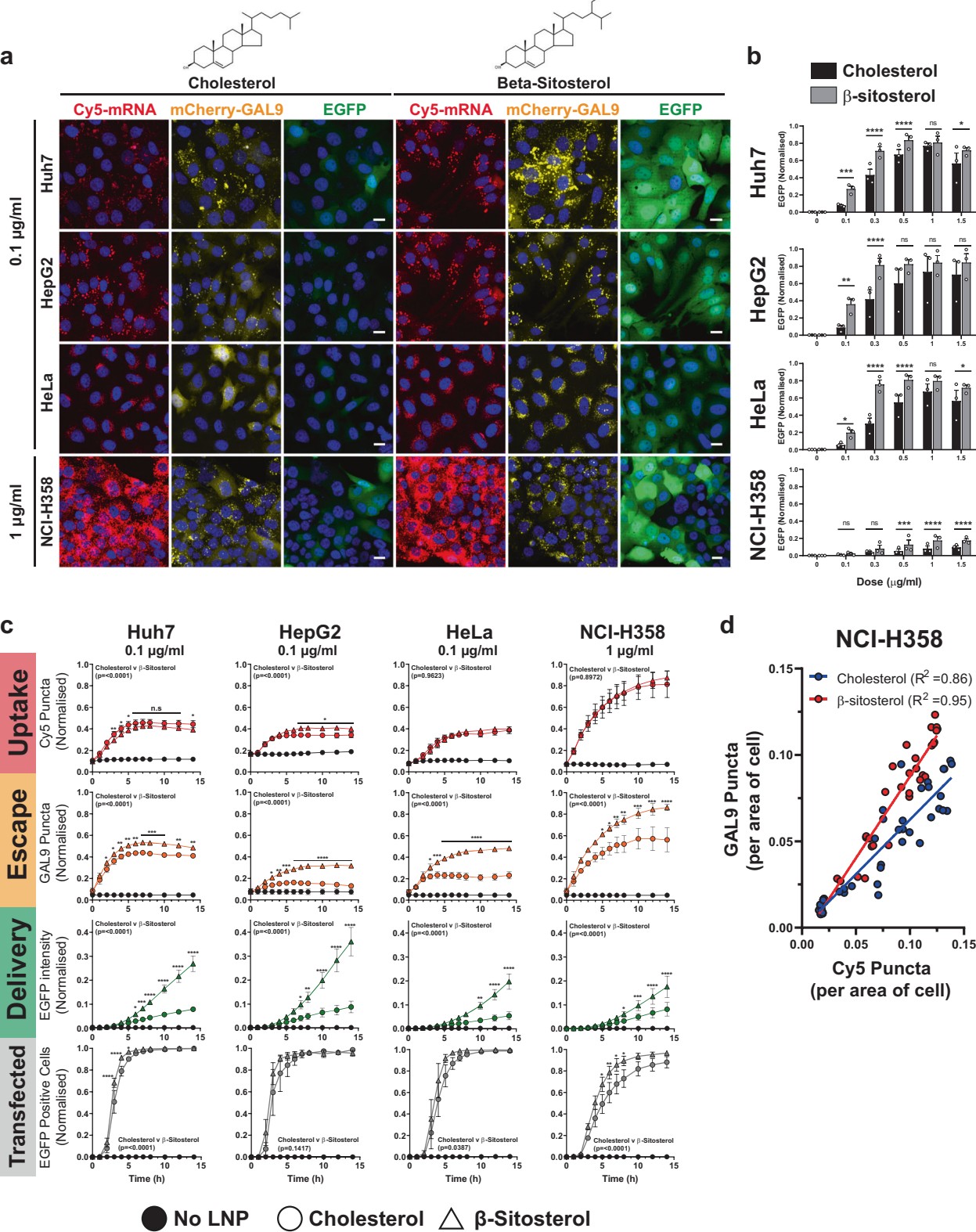

**Fig. 5 β-Sitosterol modifies endosomal escape rate. a** Huh7, HepG2, HeLa and NCI-H358 mCherry-GAL9 cells were dosed at indicated concentrations of MC3-LNPs formulated with cholesterol or β-sitosterol. Images are representative of 14 h post-dosing. Scale bar = 20 μm. **b** Comparison of cell EGFP fluorescent intensity at 14 h after incubation with Cholesterol or β-sitosterol particles in Huh7, HeLa, HepG2 and NCI-H358 cells across 0.1–1.5 μg/ml doses. Values represent normalised EGFP intensity ± SEM from n = 3 independent experiments. **c** Quantitation of (**a**) examining the formation of Cy5 positive structures, mCherry-GAL9 structures and EGFP fluorescence per cell over time. Values represent normalised means ± SEM from n = 3 independent experiments. Significance was determined in (**b**, **c**) for full 0–14 h time courses by two-way ANOVA followed by Tukey's multiple comparison test where \*p < 0.05, \*\*p < 0.01 \*\*\*p < 0.001, \*\*\*\*p < 0.0001 and ns not significant. **d** Comparison of total Cy5 or GAL9 puncta over time obtained from (**c**) for cholesterol or β-sitosterol particles in NCI-H358 cells. Linear regression carried out and R[2] values displayed on the graph.

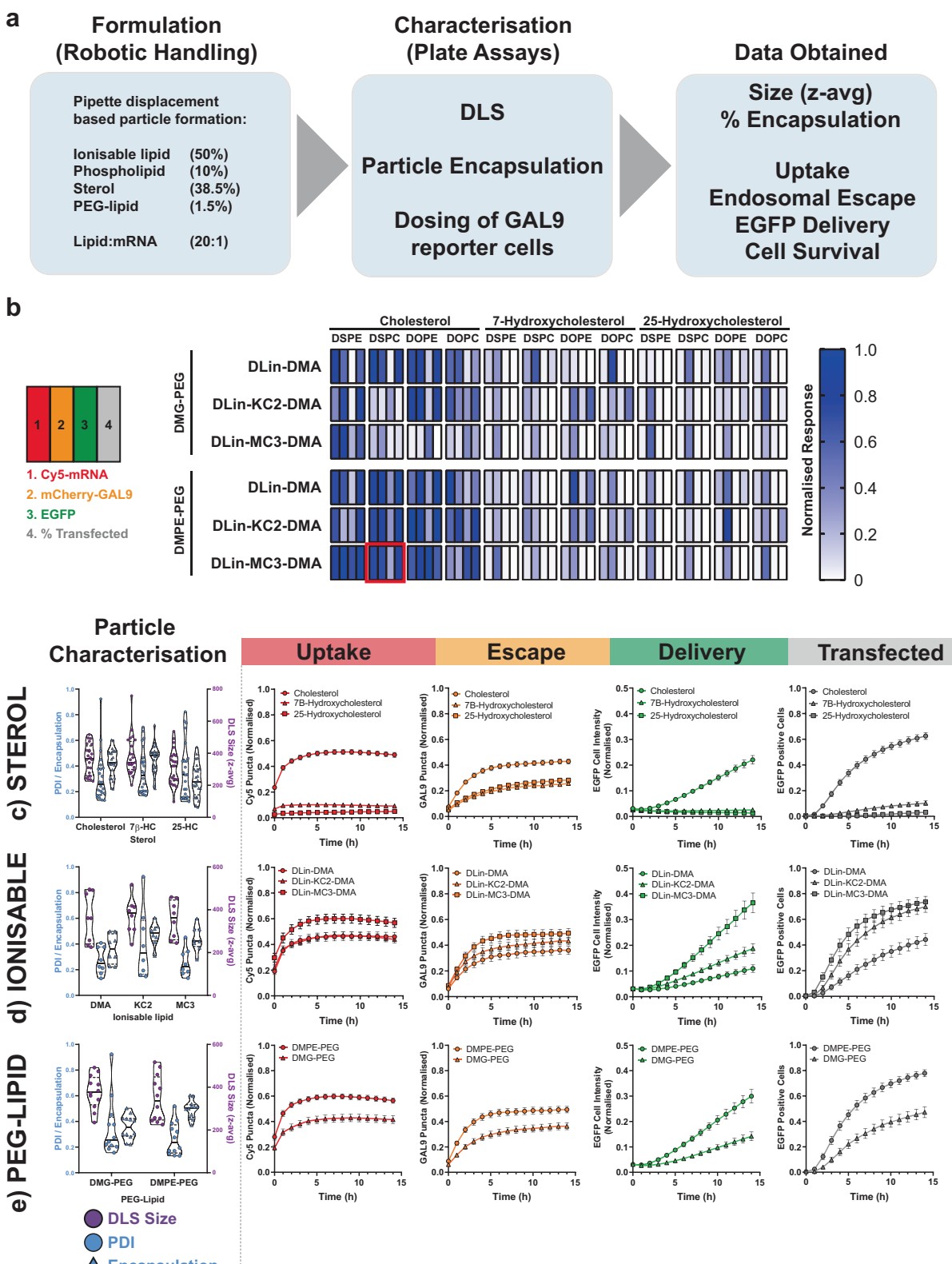

In this screen, we explored cholesterol derivatives, expanding from the analysis of β-sitosterol (Fig. 5). We substituted cholesterol to 7B-HC or 25-HC, a hydroxyl modification of cholesterol ring B or the tail region respectively. The characteristics of the 7B-HC LNPs were similar to cholesterol LNPs, however, 25-HC particles suffered from poorer encapsulation efficiencies (average 29% vs 43%, Fig. 6c—left panel). Cellular uptake of 7B-HC and 25-HC particles was severely ablated (Fig. 6b, c). Despite this, there was still recruitment of mCherry-GAL9, but interestingly this did not result in functional mRNA delivery, as evidenced by the near lack of EGFP expression and low transfection level. This suggests additional parameters affect to what extent the delivered mRNA can be utilised by the cell and may suggest the inclusion of 7B-HC or 25-HC into the LNPs

**Fig. 6 Use of mCherry-GAL9 reporter line with automated particle formulation. a** Overview of experimental approach. particles are formed by automated pipetting in a plate-based format and feed directly into plate-based assays for particle size determination by dynamic light scattering (DLS), encapsulation by ribogreen integration and for dosing mCherry-GAL9 reporter cells directly. **b** Heatmap summary of LNP formulation variation data representing normalised Cy5-mRNA uptake, mCherry-GAL9 puncta, EGFP fluorescence and % EGFP positive cell values at 14 h post-dosing. Red square indicates standard composition of particles used in rest of manuscript. **c–e** Overview of particle characteristics and kinetic results obtained with the screening approach looking at comparisons of particles based upon variable components. Values plotted are mean normalised values ± SEM **c** Sterol modified particles, $n = 24$ LNPs per group from $n = 4$ independent experiments. **d** Ionisable lipid modified, only data with cholesterol particles shown, $n = 8$ per group from $n = 4$ independent experiments **e** PEG-lipid modified, only data with cholesterol particles shown, $n = 12$ per group from $n = 4$ independent experiments.

hinders protein translation. Subsequent data comparisons, for this reason, focus solely upon LNPs containing cholesterol.

DMA and KC2 are ionisable lipids that have similar pKas of 6.7 and 6.8 respectively (~6.44 for MC3) but have been reported to have ~20-fold range in potency in mice as determined by Factor VII ED$_{50}$ (1 mg/kg (DMA) vs 0.1 mg/kg (KC2) 0.05 mg/kg (MC3))[9,42]. Consistent with this, the substitution of MC3 to DMA or KC2 led to reduced particle uptake, reduced levels of endosomal escape and consequently markedly reduced levels of EGFP expression (Fig. 6b, d). However, DMA and KC2 LNPs, whilst resulting in similar levels of uptake, perform differently with respect to their capacity to induce endosome damage (Fig. 6d), with KC2 having enhanced endosomal escape potency and thereby outperforming DMA in terms of protein expression and transfection efficiency.

Finally, we examined differences in particles containing either DMG-PEG$_{2k}$ or DMPE-PEG$_{2k}$. Whilst both contain a C14-acyl chain and PEG$_{2k}$, DMPE contains a phosphate group giving the overall molecule a negative charge compared to DMG that is neutral. Substitution to DMG-PEG$_{2k}$ resulted in reduced uptake and slower kinetics, lower endosomal escape induction and ultimately impaired EGFP production across all particles surveyed (Fig. 6b, e). It has been shown that the half-life of DMPE-PEG on nanoparticle surfaces is shorter than DMG-PEG[13] and this may account for some of the changes observed. In addition, particle charge variance may also lead to differential interactions with serum proteins that are important for cellular uptake and/or endosomal escape[13]. Representative microscopy images demonstrating changes to key particle components on the uptake, GAL9 recruitment and EGFP translation are shown in Supplementary Fig. 9.

Altogether, the screen presented in Fig. 6 illustrates that LNPs with similar physical properties such as size, PDI and encapsulation but formed from different lipids can demonstrate great variation in terms of how they are taken up by cells and are able to deliver mRNA. Integration of the mCherry-GAL9 reporter into a screening platform with multiple cellular read-outs have allowed us to probe in greater detail to which delivery step a certain lipid alteration can modulate LNP potency and highlights the benefits of utilising such a reporter system for deriving greater insights into nanoparticle function.

## Discussion

In this work, we have demonstrated a comprehensive system for the monitoring of nanoparticle trafficking from uptake, to endosomal escape, and to protein translation within a single microscopy assay and established a screening platform by integrating the assay with robotic formulation and high-throughput biophysical analysis of LNPs.

The benefits of such an assay are clear in that they enable comparison of nanoparticles across multiple relevant parameters to ascertain mechanistically why one particle formulation may outperform another, an explanation that is often absent if simply observing protein translation. By automated formulation of a

panel of 72 LNPs, we were able to examine the relative consequences of substitution of each lipid component. This approach can highlight the general effect of component substitution but reduces the reliance on using a single LNP to make the decision of efficacy. Our results indicated that a particle formulated using DLin-MC3-DMA, Cholesterol, and DMPE-PEG would perform best across uptake and endosomal escape parameters. Indeed, particles comprised of these three components and a variable phospholipid were the highest performing particles in our screen. Whilst this replicates our current understanding of well-performing LNP formulation, our study importantly demonstrates and validates that this assay can be used to survey a wider chemical space of poorly explored lipids, with a rich information read-out. This will make it possible to identify LNP formulations that outperform benchmark LNPs or can achieve the same level of delivery at lower dosing requirements.

Our analysis of substitution of cholesterol to β-sitosterol within LNPs confirms that particle uptake and endosomal escape can act independently of one another and highlights the importance of understanding and optimising both parameters during the design of next-generation LNPs. Cellular uptake of these particles was similar but endosomal escape was strongly improved by switching to β-Sitosterol, consistent with a recent report that β-sitosterol particles have a multi-faceted structure that may directly modulate endosomal fusion[40]. Even so, little is known about if there are fundamental underlying differences in the endosomal compartments that particles are taken up in or whether there are other differences in the protein corona formed that may alter particle fusion/endosomal escape properties. Interestingly, substitution to β-sitosterol did not significantly increase the maximum amount of EGFP protein attainable suggesting natural limits exist on how much exogenous mRNA can be utilised by cells. Further benefits may therefore be obtainable by modulating cellular translation rates or mRNA cargo sequences to enhance ribosome docking and translation. The relative benefit of achieving enhanced endosomal escape may not necessarily be through higher protein expression, instead the benefits may come from lower dosing requirements that thereby reduce the cost of mRNA therapies and the risk of toxic effects, in particular from the drug vehicle itself.

We were also able to induce a GAL9 response without a delivery vehicle but by using CADs such as UNC2383. The response to CADs appeared phenotypically similar to that induced by LNPs or PNPs and is consistent with the compounds proven capability to deliver oligonucleotides[29]. However, the cellular toxicity associated with these compounds tentatively suggests underlying differences in the process inducing GAL9 recruitment and may suggest a more severe endolysosomal damaging approach rather than the fusion that is anticipated with ionisable LNPs. Variation in the size of the mCherry-GAL9 structures generated by UNC2383 and chloroquine suggests underlying differences in the endolysosome membrane compartments that particles are taken up in, this may correspond with differences in utility for aiding oligonucleotide delivery. Future studies are required to evaluate whether combining CADs with

LNPs/PNPs is a viable strategy to enhance cellular delivery of mRNA, as previously shown with small oligonucleotides. Recent work has suggested that some CADs are able to induce endosomal rupture and enhance nanoparticle-mediated delivery, but this is dependent upon nanoparticle formulation[44]. In addition, the rapid and acute onset of GAL9 recruitment with UNC compounds suggests accurately synchronising compound dosing with particle uptake is likely to be critical. We have shown that mCherry-GAL9 cellular models boast a high Z' factor, making this assay amenable for subsequent high-content screening for novel CADs or LNPs that are potent inducers of endosomal rupture and that will allow us to explore this point further.

Not all delivery vehicles induce Galectin recruitment responses. Recently, extracellular vesicles were demonstrated to act in a GAL3-independent manner, however, it should be noted that GAL3 is a weaker identifier of endosomal damage than GAL9[24,45]. It will be interesting to examine whether other emerging delivery vectors for oligonucleotides such as dendrimers, particle modifications with cell-penetrating peptides or the use of viral-like particles invoke GAL9 recruitment responses that are similar to those induced by LNPs. Readout of endosomal escape with mCherry-GAL9 recruitment strongly correlates with the translation of cargo mRNA and is therefore an excellent predictor for nanoparticle efficacy. However, GAL9 recruitment is not always indicative of successful protein production as translation can still be decoupled from endosomal escape as demonstrated by the utilisation of mTOR inhibitors or the substitution of cholesterol for 7B-HC/25-HC in LNPs. Hydroxycholesterols have previously been shown to inhibit mTOR signalling and therefore protein translation[46], but could also potentially influence mRNA-lipid complexation and hinder cytosolic mRNA release.

By integrating the mCherry-GAL9 reporter into multiple cell lines, we can also begin to explore cell to cell differences in response to LNPs. Our study showed that liver-derived cell models demonstrated greater sensitivity to endosomal disruption from MC3-LNPs than non-liver cell models without equivalent differences in the level of particle uptake, providing a more detailed understanding to the observation by Sayers et al. that LNPs are differently effective in liver-derived Huh7 compared to lung-derived NCI-H358[47]. Examination of a wider range of LNP formulations and cationic lipids will be important to determine whether this is an intrinsic sensitivity of these cell models to nanoparticle delivery or whether this is indicative of a composition-driven LNP selectivity for certain cell types. Similarly, integrating the mCherry-GAL9 into a broader range of cellular models will be important for screening particle formulations for the identification of parameters important for selective uptake to ultimately facilitate selective organ targeting in vivo.

By improving our understanding of LNP characteristics that are critical to achieve particle uptake and endosomal escape across cellular models, we can ideally achieve therapeutic doses using lower amounts of potentially immunogenic lipid carriers and reduce the costs for future potential treatments. Identification of novel particle formulations will undoubtably require considerable efforts to screen nanoparticle variations in a robust manner, we believe this assay forms a fundamental basis for future screening endeavours to progress the development of drug delivery systems.

## Methods

**Materials**. Cholesterol (C8667), DOPC (P6354) and Chloroquine (C6628) were obtained from Sigma Aldrich. DSPC (LP-R4-076), DOPE (LP-R4-069) were from Merck-Millipore. 7β-hydroxycholesterol (700035P), 25-hydroxycholesterol (700019P), β-sitosterol (700095P) and DSPE (850715P) were from Avanti Polar Lipids. DMG-PEG$_{2k}$ (GM-020) and DMPE-PEG$_{2k}$ (PM-020CN) were from Nof America Corporation. Cationic lipids DLin-DMA, DLin-KC2-DMA, DLin-MC3-DMA and small molecules KU0063794, Rapamycin, Torin1, UNC2383, UNC4167, UNC10217938A were chemically synthesised.

**Cell line maintenance and generation of GAL9 reporter**. HeLa (CCL-2), HepG2 (HB-8065) and NCI-H358 (CRL-5807) cells were purchased from ATCC whilst Huh7 (Riken - RCB1366) were a kind gift from Prof. Samir El-Andaloussi (KI, Stockholm), all cell lines were authenticated by STR profiling and tested negative for mycoplasma contamination. Cells were maintained at 37 °C in a humidified incubator in a complete media of DMEM + Glutamax (Huh7, HeLa, HepG2) or RPMI + Glutamax (NCI-H358 cells) both supplemented with 10% foetal bovine serum.

Stable cells expressing mCherry-GAL9 were generated by knock-in at the AAVS1 locus. Cells were seeded at $2 \times 10^5$ cells/well (12-well) and transfected with mCherry-GAL9 reporter:AAVS1 zinc-finger nuclease (1:9) using FuGENE HD transfection reagent (Promega) as per manufacturer's instructions. Cells were incubated for 48 h before the addition of 1 µg/ml Puromycin to select for stably integrated cells. Stable cells were subsequently sorted into similar expression pools by flow cytometry.

**Imaging experiments and quantitation**. Cells were seeded into 384-well Cell-Carrier Ultra plates (PerkinElmer: 6007558) in complete media a minimum of 16 h prior to treatment. Hoechst 33342 was added to cell culture medium at 0.5 µg/ml a minimum of 1 h prior to imaging experiments to evenly stain nuclei prior to experiment start points. Nanoparticle or small molecules were first dispensed into a 384-well source plate (Greiner: 781280) containing appropriate complete media + 0.5 µg/ml Hoechst by using an Echo 655T acoustic dispenser (Labcyte). At the experimental start point, the media on cells was removed and replaced directly by media containing LNPs/small molecules from the source plate using a Bravo liquid handling robot (Agilent). The cell plate was then moved to the microscope and imaged immediately. Live-cell experiments were carried out within a humidified imaging chamber maintained at 5% CO$_2$ with a CV7000 (Yokogawa) spinning disk confocal microscope utilising a 20x objective (NA 0.75). Images were obtained using a 405 nm laser (BP445/45 nm), 488 nm laser (BP522/35), 561 nm laser (BP600/37) or 640 nm laser (BP676/29) for relevant fluorophores. For microscopy time-course measurements, the same fields of view were imaged over time that had received the experimental treatment noted in the figure. Images were processed and analysed for relevant features and parameters indicated in figures utilising Columbus image-analysis software (Perkin Elmer, v2.9.0). Briefly, spots were identified using maximum intensity projection fluorescence images. Cells were identified using Hoechst 33342 for nuclei detection and combined mCherry-GAL9 and EGFP signal (where appropriate) for individual cell boundaries. Within individual cell regions of interest, spot populations were quantified using a 'Find spots' building block within Columbus software that identifies punctate structures with relative intensities higher than local background cellular intensity, no limits were placed on max puncta size. Data were exported and handled in Spotfire (Tibco, v10.3), in many cases numerical values obtained were normalised for combining between experimental replicates. Data were exported and plotted with Prism (Graphpad, v8.0.1).

**PCR and droplet digital PCR**. DNA was extracted from wild-type (WT) and GAL9 transgenic cell pools using QuickExtract™ DNA extraction solution (Lucigen), according to the manufacturer's instructions. Briefly, extraction solution was added to cell pellets, followed by samples vortexing and incubation at 65 °C for 6 min. Samples were vortexed again and incubated at 98 °C for 2 min. For ddPCR, the same amount of DNA (50 ng) for all samples was mixed with ddPCR supermix for probes, puromycin-FAM primers/probe mix, AP3B1-HEX primers/probe mix (all from Bio-Rad), and Hind III 5U (New England BioLabs Inc.) restriction enzyme. Samples were incubated for 10 min at room temperature (RT) for DNA fragmentation. PCR droplets were generated on an Automatic Droplet Generator using oil for probes (all from Bio-Rad). Target genes were amplified on a C1000 thermal cycler (Bio-Rad) according to the following conditions: 95 °C for 10 min, followed by 40 cycles of template denaturation at 94 °C for 30 s, and annealing and extension at 57 °C for 1 min, with a temperature ramping ratio of 2 °C/s, with a final incubation step at 98 °C for 10 min. PCR droplets were analysed on a QX200 droplet reader equipped with QuantaSoft software v.1.7.4 (Bio-Rad). Populations of fluorescent droplets were identified as depicted in Supplementary Fig. 2 and the average copy number of target gene insertions per cell was calculated using the Copy Number Variation algorithm, considering a 2x ploidy for HepG2 and Huh7 cell lines, and a 3x ploidy for HeLa and NCI-H358 cell lines, and normalised to the copy number of the reference gene AP3B1. For endpoint PCR, DNA was first further purified by mixing with ice-cold ethanol 99,9% in a ratio 1:3 (v/v), followed by overnight incubation at −20 °C, and then by washing with ethanol 70%. Samples were dried at RT and DNA dissolved in nuclease-free water. Purified DNA (100 ng) was then mixed with Phusion Flash PCR master mix (ThermoFisher Scientific) and primers spanning the AAVS1 locus and the donor construct, followed by incubation on a C1000 thermal cycler (Bio-Rad) according to the following conditions: 98 °C for 10 sec, followed by 35 cycles of template denaturation

at 98 °C for 1 s, annealing at 54 °C for 5 s and extension at 72 °C for 15 s, with a final incubation step at 72 °C for 1 min. Amplicons were then resolved in a 2% agarose E-gel, using TrackiT 1 Kb plus DNA ladder (all from ThermoFisher Scientific), followed by gel imaging on a Gel-Doc system (Bio-Rad).

**Fluorescence-activated cell sorting (FACS)**. WT and mCherry-GAL9 reporter cells were grown and expanded until 80% confluency. Cells were washed twice with PBS, detached with TrypLE Express (10 min, 37 °C), resuspended in FACS buffer (5% FBS in PBS) and filtered in FACS tubes using a 35 μm mesh (Falcon). Cell sorting was performed on a FACSAria™ III cytometer (from BD Instrument), using the BD FACSDiva Software. For each cell type, WT cells were used to set up the voltage of the forward scatter (FSC) and side scatter (SSC) as well as to adjust the autofluorescence level in the mCherry channel. Debris was excluded by plotting the FSC vs SSC with all events and gating for living cells, and cell aggregates (doublets/clusters) were excluded by plotting the SSC-height vs SSC-area with living cells and gating for single cells. mCherry-positive GAL9 reporter cells were sorted in bulk according to two fluorescence intensity levels. Following sorting, cells were spun down (200 g, 5 min at RT) to remove excess FACS buffer and then grown for a week in presence of 1% penicillin-streptomycin. Antibiotics-free culture medium was used for cell experiments.

**Western blotting**. Cell pellets ($5 \times 10^6$ cells) were lysed in RIPA buffer (Thermo Fisher Scientific) supplemented with protease inhibitors (Sigma-Aldrich) for 15 min on ice. Total protein content was determined using the Qubit protein assay kit (Thermo Fisher Scientific) following the manufacturer´s protocol. Protein lysates (40 μg/lane) were separated by SDS-PAGE and transferred to polyvinylidene fluoride membranes (BioRad). Membranes were blocked with Intercept (TBS) blocking buffer (LI-COR) for 1 h at RT and incubated with primary antibodies mCherry (E5DF8, #43590) and β-Tubulin (9F3, #2128) (Cell Signalling Technologies) diluted 1:1000 in blocking buffer at 4 °C overnight. Membranes were washed three times with 0.1% TBS-Tween and incubated for 1 h at RT with IRDye 800CW goat anti-rabbit IgG cat# 926–32211 (LI-COR) diluted 1:20,000 in 0.1% TBS-Tween. Following washes, membranes were visualised with the Odyssey CLx imaging system and processed in Image Studio (v4.0, LI-COR).

**Formation of LNPs (NanoAssemblr)**. DLin-MC3-DMA cholesterol or β-sitosterol containing LNPs were formulated using a NanoAssemblr (Precision NanoSystems) by microfluidic mixing chip. Briefly, lipids were prepared in Ethanol at a ratio of 50:38.5:10:1.5 (MC3:Sterol:DSPC:DMPE-PEG2000) at a 10:1 (w/w) Lipid:mRNA cargo, N:P ratio = ~3:1. mRNA cargo encoding for EGFP was prepared at a 4:1 ratio (Unlabelled:Cy5 labelled – TriLink: L7201/7701) in 50 mM Citrate pH 3 buffer (TekNova: Q2445). Lipid and mRNA containing solutions were mixed 3:1 (Citrate:Ethanol) at a constant flow rate of 12 ml/min to form nanoparticles. Particles were dialysed overnight into PBS pH 7.4 at 4 °C and sterile filtered using a 0.23 μm filter. Characteristics of LNP batches used in this study are shown in Supplementary Table 2.

**Formation of LNPs (automated pipetting)**. Robotic formulation of particles was achieved utilising a Bravo liquid handling robot utilising VWorks software (v12.2.0.1306, Agilent). Particles were prepared at a ratio of 50:38.5:10:1.5 (Cationic Lipid:Sterol:Phospholipid:PEG-Lipid) at a 20:1 (w/w) lipid:mRNA cargo, N:P ratio = ~6:1. 7.5 μl of each lipid component was combined per well to a final volume of 30 μl. mRNA cargo encoding for EGFP was prepared at a 4:1 ratio (Unlabelled:Cy5 labelled – TriLink: L7201/7701) in 50 mM Citrate pH 3 buffer (TekNova: Q2445) at 30 μl per well (122 μg/ml). 10 μl of the lipid-ethanol mix was added directly to mRNA mix and pipetted up and down 10 × 20 μl. An equal volume (40 μl) of PBS pH 7.4 was then added to the mRNA/lipid and mixed a further 5 × 20 μl and then incubated at 4 °C overnight prior to usage.

**Formation of PNPs (automated pipetting)**. Polyplexes were prepared utilising a Bravo robot as for LNPs (above). 20 μl of 50 mM citrate pH 3 (TekNova #Q2445) buffered mRNA (4:1 GFP:Cy5 –TriLink CleanCap mRNA L-7201/L-7701) (100 μg/ml) was injected into wells (Greiner V-bottom #781280) that contain 20 μL of the desired polymer solution at an amine:phosphate ratio of 8:1 assuming 50% amines charged per polymer. After addition of the RNA solution, polyplex suspensions were mixed through 10 × 15 μl mix steps and incubated at RT for 30 min prior to addition of an equal volume (40 μl) 1 M Tris-HCl pH 8 (TekNova #T1080). Polyplexes were mixed 5 × 15 μl before moving to 4 °C for 16 h prior to use.

**Particle characterisation – DLS and encapsulation**. Size (Diameter, Z-average) of particles and polydispersity index (PDI) was determined by dynamic light scattering utilising a Malvern Zetasizer ($\lambda = 633$ nm, scattering angle = 173°) for particles formed by NanoAssemblr or a Malvern Zetasizer APS ($\lambda = 832$ nm, scattering angle = 90°) for particles formed by pipetting. In both cases standard viscosity and refractive index values for pure water at 25 °C, 0.8872 mPa and 1.33 respectively, were used for data analysis within Zetasizer software (v7.12, Malvern).

Ribogreen dye (Thermo Fisher Scientific) was used according to manufacturer's guidelines ± 1% Triton to ascertain encapsulated mRNA by comparison to a relevant mRNA standard curve.

**Statistics and reproducibility**. Statistical testing was carried out in Prism (Graphpad, v8.0.1) with relevant multiple comparisons and post-test where appropriate. No formal sample-size calculation was performed and no randomisation strategy was applied. For microscopy experiments, data were collected for as many fields of view as possible whilst maintaining time-points required for time-course experiments. Technical duplicates within each experiment were obtained as a minimum. No data were excluded from the analysis and all reported results were replicable. Statistical testing was carried out upon a minimum of three independent experimental replicates. See figure legends for full details of replicates, statistical testing and significance.

**Reporting summary**. Further information on research design is available in the Nature Research Reporting Summary linked to this article.

## Data availability
The source data underlying all quantitative figures are provided as Supplementary Data 1. All data supporting the findings of this study are available from the corresponding author upon reasonable request.

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

## Acknowledgements

M.J.M. and A.M.S. are PostDoc fellows of the AstraZeneca R&D PostDoc Program. G.O. is a member of the AstraZeneca R&D graduate programme. This work was conducted within the Industrial Research Centre, FoRmulaEx–Nucleotide Functional Drug Delivery, and with associated financial support to E.K.E. from the Swedish Foundation for Strategic Research (SSF, grant No. IRC15-0065). We thank Erik Oude Blenke for the formulation of β-sitosterol containing LNPs.

## Author contributions

M.J.M. and G.O. performed experiments. Study design and critical discussions were carried out between M.J.M., G.O., A.C., E.K.E. and A.S. PCR experiments were carried out by A.M.S., western blots by E.L.I. and FACS of reporter lines was performed by A.G. Polymers were generated by J.T.W. The manuscript was written by M.J.M., E.K.E. and A.S. with input from all authors.

## Competing interests

The authors declare the following competing interests: M.J.M., G.O., A.M.S., E.L.I., A.G., A.C. and A.S. are or were employees of AstraZeneca plc. All other authors declare no competing interests.
