## [Transparent Peer Review File · Communications Biology]

Reviewers' comments:

Reviewer #1 (Remarks to the Author):

This paper by Munson et al, is a method building paper largely based off of work by Anders Wittrup using Galectin-9 as a detector for endosomal escape. Using 4 different stable cell lines expressing mCherry-Gal9, the authors used a battery of tests to determine if the cargo (Cy5-mRNA expressing eGFP) escaped the endosome with different lipid nanoparticles carrier compositions. Overall, the authors did an excellent job of presenting the work showing the expression levels of mCherry-Gal9 in their cell lines, demonstrating that endosome disruptors such as chloroquine caused mCherry-Gal9 puncta to form and that the cargo was escaping through eGFP expression in a dose and time dependent manner according to the conditions presented. The manuscript demonstrates all of the required controls that are necessary for assessing transfection efficiency, cargo loading in LNPs, LNP compositions, and vesicle tracking. I could not find any scientific fundamental flaws and thought that this is beautiful work.

The only issue that gives me pause is that this paper, and other papers before it (as far as I have been able to find) do not show that disruption of the CRD domain of Gal9 disperses the puncta formation when stimulated with chloroquine or some other endosome disruptor. Galectins seem to do just about everything in biological terms. Disruption of the mCherry-Gal9 will not affect cargo escape due to the redundancy and expression of endogenous Galectins, but may prove that Gal9 is binding to membrane β -galactosides and directly involved with endosomal escape of cargo and not through an intermediary.

One minor issue: Page 3, line 54 is not necessarily true as some of the SARS-CoV-2 vaccines are mRNA based and have passed phase 3 clinical trials, FDA approval and will be on the market very shortly.

Reviewer #2 (Remarks to the Author):

This is a highly interesting and well written paper. It uses the phenomenon that galectin-9 (like other galectins) accumulate around disrupted intracellular vesicles. Analysis of this is built into a strategy with other assays aimed at following and improving uptake and release into cytosol of therapeutic RNA and similar. Only one aspect needs to be improved:

How are the galectin-9 dots counted? The methods give software used but not any detail of considerations on how to define and count the puncta. How big are they set to be? Were they counted in one cell plane or several, separately or at the same time? Were there other criteria what should be regarded as puncta compared to background aggregates. Etc. This consideration applies also to other puncta counting assays in the paper, but as the galectin-9 puncta is the main topic, the question is about that.

A minor issue for discussion. The peak effects seem to occur after 1 or more hours of incubation with agents. However, it is known that galectins accumulate around a vesicle within seconds after disruption, e.g. ref 22. Endocytosis itself occurs within minutes, and the endocytic pathway from cell surface to lysosome takes no more than 20 minutes. So, what is it that takes the extra time?

Reviewer #1 (Remarks to the Author):

This paper by Munson et al, is a method building paper largely based off of work by Anders Wittrup using Galectin-9 as a detector for endosomal escape. Using 4 different stable cell lines expressing mCherry-Gal9, the authors used a battery of tests to determine if the cargo (Cy5-mRNA expressing eGFP) escaped the endosome with different lipid nanoparticles carrier compositions. Overall, the authors did an excellent job of presenting the work showing the expression levels of mCherry-Gal9 in their cell lines, demonstrating that endosome disruptors such as chloroquine caused mCherry-Gal9 puncta to form and that the cargo was escaping through eGFP expression in a dose and time dependent manner according to the conditions presented. The manuscript demonstrates all of the required controls that are necessary for assessing transfection efficiency, cargo loading in LNPs, LNP compositions, and vesicle tracking. I could not find any scientific fundamental flaws and thought that this is beautiful work.

We thank the reviewer for their very kind words and appreciate that our attempts to provide a comprehensive overview of the GAL9 assay system and applications has been recognised.

The only issue that gives me pause is that this paper, and other papers before it (as far as I have been able to find) do not show that disruption of the CRD domain of Gal9 disperses the puncta formation when stimulated with chloroquine or some other endosome disruptor. Galectins seem to do just about everything in biological terms. Disruption of the mCherry-Gal9 will not affect cargo escape due to the redundancy and expression of endogenous Galectins, but may prove that Gal9 is binding to membrane β -galactosides and directly involved with endosomal escape of cargo and not through an intermediary.

We agree with the reviewer that it is a very interesting point to further understand the biological basis for the GAL9 recruitment to damaged endosomal structures.

It has been shown that a GAL3 construct containing an R186S mutation in the carbohydrate binding domain reduced lectin binding affinity by ~40-fold^{1,2}. Furthermore, GAL3 wild-type protein is recruited to endosomes damaged by invading salmonella bacteria during endosomal escape, but GAL3 R186S fails to localise to the bacterial endosomes³. This suggests that at least GAL3 functions via its CRD domain, with Galectin family proteins containing highly conserved CRD domains⁴. We refer to this work in the introduction but have made the mechanistic connection clearer in the revised text (Lines 69-72)

Members of the Galectin (GAL/LGALS) family of proteins have been exploited as reporters of endosomal escape in a variety of contexts. Galectins are primarily expressed in the cytosol and contain carbohydrate recognition domains (CRDs) that bind to β -galactoside sugars¹⁷. Galectins can be recruited to endosomes when membrane damage exposes β -galactosides on the inner leaflet of the endosomal membrane to the cytosol¹⁸. Point mutations that strongly reduce the binding affinity of the CRD also prevented endosomal relocalisation of GAL3¹⁸⁻²⁰.

Assuming that endosomal escape induced by bacteria is inherently similar to that of small molecules/LNPs, we would infer that the carbohydrate recognition domains (CRDs) of Galectins are also directly required for recruitment to endosomal membranes. CRDs are highly conserved in terms of structure

Given that our major focus being was on the reagents, methodology and analysis necessary to comprehensively evaluate the efficacy and function or delivery systems, we feel experiments

examining the mechanistic biology of Galectin proteins is beyond the scope of our work. However we hope that the reviewer will still appreciate the references above and the likely cross-over/applicability to chloroquine or LNP-induced endosomal escape.

One minor issue: Page 3, line 54 is not necessarily true as some of the SARS-CoV-2 vaccines are mRNA based and have passed phase 3 clinical trials, FDA approval and will be on the market very shortly.

We completely agree with the reviewer that the information we entered here no longer holds true in what is a rapidly developing and exciting area of progress. We have removed that part of the text and updated to the following (Line 52-54) to highlight endosomal escape still limits efficacy of oligonucleotide products.

“Improving endosomal escape will be of great importance for the further development of oligonucleotide therapeutics, particularly for treatments that require high or sustained therapeutic protein levels”

Reviewer #2 (Remarks to the Author):

This is a highly interesting and well written paper. It uses the phenomenon that galectin-9 (like other galectins) accumulate around disrupted intracellular vesicles. Analysis of this is built into to a strategy with other assays aimed at following and improving uptake and release into cytosol of therapeutic RNA and similar.

We thank the reviewer and appreciate the interest and enthusiasm for our work.

Only one aspect needs to be improved:

How are the galectin-9 dots counted? The methods give software used but not any detail of considerations on how to define and count the puncta. How big are they set to be? Were the counted in one cell plane or several, separately or at the same time? Were there other criteria what should be regarded as puncta compared to background aggregates. Etc. This consideration applies also to other puncta counting assays in the paper, but as the galectin-9 puncta is the main topic, the question is about that.

We apologise for the limited information provided surrounding the spot counting methodology utilised within the paper and are grateful to the reviewer for pointing this out.

Spots were identified using the Columbus image analysis program on maximum intensity projections of fluorescence images. Cells were identified using Hoechst 33342 for nuclei detection and combined with mCherry-GAL9 and EGFP fluorescence (where appropriate) for identifying individual cells. Within individual cell regions of interest, GAL9+ spots (or other spot populations) were determined using a ‘Find spots’ building block within Columbus software that identifies objects with a relative intensity higher than local background cellular intensity. No limits were placed on max puncta size. Background GAL9+ structures, aggregates or auto fluorescent structures that passed these criteria were still included in the analysis, with the no treatment control used to represent the level of background structures identified (e.g HepG2 cells in Fig. 3d, the black no LNP control is higher than other cell lines due to auto fluorescent structures that may represent lipid droplets). We have therefore updated the paper to include the following in the methods section (Lines 480-485):

“Images were processed and analysed for relevant features and parameters indicated in figures utilising Columbus image-analysis software (Perkin Elmer, v2.9.0). Briefly, spots were identified using maximum intensity projection fluorescence images. Cells were identified using Hoechst 33342 nuclei detection and combined mCherry-GAL9 and EGFP signal (where appropriate) for individual cell boundaries. Within individual cell regions of interest, spot populations were quantified using a ‘Find spots’ building block within Columbus software that identifies punctate structures with relative intensities higher than local background cellular intensity, no limits were placed on max puncta size.”

A minor issue for discussion. The peak effects seem to occur after 1 or more hours of incubation with agents. However, it is known that galectins accumulate around a vesicle within seconds after disruption, e.g. ref 22. Endocytosis itself occurs within minutes, and the endocytic pathway from cell surface to lysosome takes no more than 20 minutes. So, what is it that takes the extra time?

We agree with the perceptive observations of the reviewer that in theory endocytosis and lysosomal delivery occurs quickly along with rapid Galectin recruitment in the case of endosomal escape, however in the case of lipid nanoparticles there are additional factors that delay and prevent immediate particle uptake and the subsequent GAL9 response. Notably, ionisable LNP uptake by cells is assisted by the formation of a protein corona that can act as a natural ligand to aid in receptor-mediated uptake of LNPs, such as ApoE adsorption onto MC3 particles⁵. Furthermore, PEG-Lipids act to ‘shield’ particles in circulation and cell culture media. Removal of PEG (PEG shedding) or occlusion of the PEG molecules (possibly through development of a protein corona) is therefore required before efficient cellular uptake of LNPs is observed and accounts for the delayed uptake.

We have noted this point in the manuscript (Lines 178-181):

“We observed a delayed onset and peak of Cy5 and GAL9 puncta formation between 4h-10h in cell lines following LNP addition and this likely represents the time required for PEG-Shedding and/or protein corona formation in the cell culture media. These changes are prerequisites for efficient particle uptake.”

It is however possible to observe with polymer nanoparticles (Fig. 4e) that after a single PNP is internalised at 0.5h, it becomes GAL9 positive at a timepoint between 0.75-1h (15-30 minutes after uptake), this would fit closely within the quoted endocytosis trafficking time period of ~20 minutes stated by the reviewer.

1. Delacour, D. *et al.* Apical sorting by galectin-3-dependent glycoprotein clustering. *Traffic* **8**, 379–388 (2007).
2. Cumpstey, I., Salomonsson, E., Sundin, A., Leffler, H. & Nilsson, U. J. Studies of arginine-arene interactions through synthesis and evaluation of a series of galectin-binding aromatic lactose esters. *ChemBioChem* **8**, 1389–1398 (2007).
3. Paz, I. *et al.* Galectin-3, a marker for vacuole lysis by invasive pathogens. *Cell. Microbiol.* **12**, 530–544 (2010).
4. Modenutti, C. P., Capurro, J. I. B., Lella, S. Di & Martí, M. A. The Structural Biology of Galectin-Ligand Recognition : Current Advances in Modeling Tools , Protein Engineering , and Inhibitor Design. **7**, (2019).
5. Akinc, A. *et al.* Targeted delivery of RNAi therapeutics with endogenous and exogenous ligand-based mechanisms. *Mol. Ther.* **18**, 1357–64 (2010).

REVIEWERS' COMMENTS:

Reviewer #1 (Remarks to the Author):

The authors address my concerns and made appropriate revisions. I think that this manuscript is suitable for publication.

Reviewer #2 (Remarks to the Author):

The authors have responded very well to critique of original submission.